# Comparison of lamina cribrosa properties and the peripapillary vessel density between branch retinal vein occlusion and normal-tension glaucoma

**Je Moon Woo[1], Jae Bong Cha[1], Chang Kyu Lee[1,2]***

**1** Department of Ophthalmology, Ulsan University Hospital, University of Ulsan College of Medicine, Ulsan, South Korea, **2** Biomedical Research Center, Ulsan University Hospital, University of Ulsan College of Medicine, Ulsan, South Korea

\* coolleo@uuh.ulsan.kr

## Abstract

### Purpose

To compare the properties of the lamina cribrosa (LC) and the peripapillary vessel density between branch retinal vein occlusion (BRVO) and normal-tension glaucoma (NTG), using swept-source optical coherence tomography and optical coherence tomography angiography.

### Methods

This retrospective study included 21 eyes of 21 patients with BRVO and 43 eyes of 43 patients with NTG who were treated from June 2016 to September 2017. The anterior LC depth (ALCD) and LC thickness (LCT) at the mid-superior, central, and mid-inferior levels; the mean difference in ALCD; and the peripapillary vessel density in the superficial and deep capillary plexuses and the choriocapillaris were compared between groups.

### Results

ALCD at the mid-superior, central, and mid-inferior levels was significantly greater in the NTG group (P < 0.05), while LCT was comparable between the groups. The mean difference in ALCD was significantly greater in the BRVO group (P = 0.03). The peripapillary vessel density in the superotemporal segment of the superficial capillary plexus was significantly lower in the BRVO group, while the density in all segments of the choriocapillaris was significantly lower in the NTG group (P < 0.05 for all).

### Conclusions

Our findings demonstrate that BRVO and NTG have different LC structures and peripapillary vessel densities.

**Data Availability Statement:** Data are available from ulsan university hospital ethics committee (contact via 0716782@uuh.ulsan.kr/Ms. Jeong

Sook Kim; secretary of ulsan university hospital institutional review board).

**Funding:** This research was supported by Basic Science Research Program through the National Research Foundation of Korea(NRF) funded by the Ministry of Science and ICT(NRF-2017R1C1B5018279).

**Competing interests:** No authors have competing interests.

## Introduction

Retinal vein occlusion is a common cause of vision loss after diabetic retinopathy and age-related macular degeneration [1, 2]. It has two forms: central retinal vein occlusion and branch retinal vein occlusion (BRVO). BRVO is associated with a higher prevalence of arterial hypertension, peripheral vascular disease, venous disease, and peptic ulcer [3]. Some patients with BRVO have clinical signs, such as optic disc hemorrhage, an increased cup-to-disc ratio, and visual field defects, that are commonly found in glaucoma patients [4]. This suggests that BRVO and glaucoma, particularly normal-tension glaucoma (NTG), may have a similar pathological mechanism such as blood flow insufficiency, although NTG is more often associated with status of hypotension, headache and generalized vasoconstriction of blood vessels especially on the extremities [5–8]. Moreover, some cases of chronic BRVO are difficult to distinguish from those of NTG because of the similar topographical and morphological characteristics [8, 9]. However, the clinical prognoses of BRVO and NTG are different. While BRVO remains stable, unless there is a recurrence of retinal embolism, NTG is a progressive optic neuropathy.

Accordingly, comparisons of the ocular regions primarily affected in these two conditions, i.e., the lamina cribrosa (LC) and peripapillary vessels [10, 11], may be key to differentiating these conditions and may provide insight into the pathogenesis of NTG. To our knowledge, no study to date has performed such comparisons because of the low penetration and detection abilities of conventional devices and the prohibitive invasiveness of existing equipment required for the evaluation of these sites. However, the availability of swept-source (SS) optical coherence tomography (OCT), one of the latest noninvasive, high-speed imaging techniques, using longer wavelengths, has increased in recent years. This technique allows better imaging of posterior eye structures, such as the LC, than does enhanced-depth imaging OCT [12]. OCT angiography (OCTA) is another new imaging technique that allows visualization of all layers of the retinal and choroidal microvasculature, without requiring dye injection [13].

The aim of the present study was to evaluate the differences in the properties of the LC and the peripapillary vessel density between BRVO and NTG, using SS-OCT and OCTA, in order to provide a deeper understanding of the pathogenesis of NTG.

## Materials and methods

### Subjects

This retrospective cohort study adhered to the tenets of the Declaration of Helsinki and was approved by the Ulsan University Hospital Institutional Review Board (UUH 2017-09-037-001). All subjects were enrolled between June 2017 and September 2018. Our IRB committee waived the requirement for informed consent, due to the retrospective nature of the study and the full anonymity of the data before we accessed them.

Patients with NTG were recruited from the Department of Ophthalmology at Ulsan University Hospital, South Korea, using the following inclusion criteria: availability of an average of three intraocular pressure measurements obtained using Goldmann applanation tonometry at different time points, $< 21$ mmHg; cup-to-disc ratio for each eye of $\geq 0.5$ or a difference between the two eyes of $> 0.2$; a defect in the retinal nerve fiber layer (consistent with a glaucomatous change in the optic nerve) observed on a fundus photograph taken under red-free light or on an optical coherence tomography image; evidence of glaucomatous visual field loss observed using a Humphrey Field Analyzer (with the Swedish Interactive Threshold Algorithm-Standard 30–2 or 24–2 program); and confirmation of open-angle glaucoma by gonioscopic examination or the Van Herick method.

NTG eyes were selected by the following criteria: 1) eyes of patients with unilateral NTG; 2) in case of bilateral NTG, the eye with higher resolution OCT image; and 3) in case of similar resolutions of the OCT images in patients with bilateral NTG, the right eye was selected to minimize selection bias.

Age-matched patients with BRVO were enrolled from the Retinal Clinic at Ulsan University Hospital, South Korea. The presence of BRVO was determined based on ophthalmoscopic slit-lamp fundus examinations, color fundus photography and OCT tomography [14]. BRVO was identified when a localized area of the retina exhibited scattered superficial and deep retinal hemorrhage, venous dilation, intraretinal microvascular abnormalities, and occluded and sheathed retinal venules. Additional inclusion criteria were as follows: intraocular pressure < 21 mmHg; minimum 6-month follow-up for stabilizing the retinal nerve fiber layer thickness [15]; and no laser therapy during the follow-up period, in order to exclude the effects of this therapy on the retinal nerve fiber layer and peripapillary vessels.

The exclusion criteria were as follows: primary open-angle glaucoma diagnosed before or during the follow-up period; best-corrected visual acuity < 0.1 on a decimal scale or > 1.0 on the logarithm of the minimum angle of resolution scale; presence of uveitis or other retinal diseases; laser therapy or intraocular surgeries, such as cataract surgery or vitrectomy, during the follow-up period; myopia of ≥ −6 D; and a history of cerebrovascular disease, ocular trauma, or any other condition that could affect the retinal nerve fiber layer and LC.

## SS-OCT imaging of the optic nerve head

The optic nerve in each patient was imaged using a commercially available SS-OCT device (DRI OCT Triton, Topcon, Japan) with a wavelength of 1050 nm and a swept source with a tuning range of 100 nm for optic nerve head scanning. The device produces scans with a axial resolution of 8 μm and a transverse resolution of 20 μm. This instrument provides high-quality images with constant signal strength throughout the posterior pole, even in eyes with cataracts that affect image quality [16]. Moreover, the instrument performs up to 100,000 A-scans per second and has an invisible scanning line because of the long wavelength; this can reduce eye movement and ensure more successful scans. We obtained images using a 6 × 6-mm area with a depth of 2.6 mm centered on the optic nerve head. Each dataset comprised 256 cross-sectional B-scan images of 512 × 256 pixels. For inclusion in the analyses, all images were required to have an image quality score of ≥ 35 according to the manufacturer's recommendations.

For identification of the anterior and posterior borders of the LC, horizontal SS OCT B-scans were assessed using Adobe Photoshop CS2 (version 9.0; Adobe Systems, Inc., San Jose, California, USA). The LC thickness (LCT) was defined as the distance between the anterior and posterior borders of the highly reflective region seen beneath the optic nerve head on a cross-sectional B-scan image (Fig 1).

LCT was determined using a method similar to that of Kwun et al. [17], in which measurements were obtained from three lines, i.e., the mid-superior, central, and mid-inferior lines of the optic disc, using the manual caliper tool in DRI OCT Viewer 9.01 software. The mid-superior and mid-inferior lines were identified as the horizontal line located at the halfway point on the vertical line that connects the optic disc center to the margin. At each line, LCT was measured at three locations along a perpendicular line from the reference line connecting the two ends of the Bruch membrane, i.e., at the center of the reference line, and 100-μm temporally and 100-μm nasally from this line. The center of the reference line was used for comparison between the two diseases, and the nasal and temporal lines were used as supporting values for the central reference line. Anterior LC depth (ALCD) was defined as the distance between the reference line connecting the end of the Bruch membrane and the anterior border of the

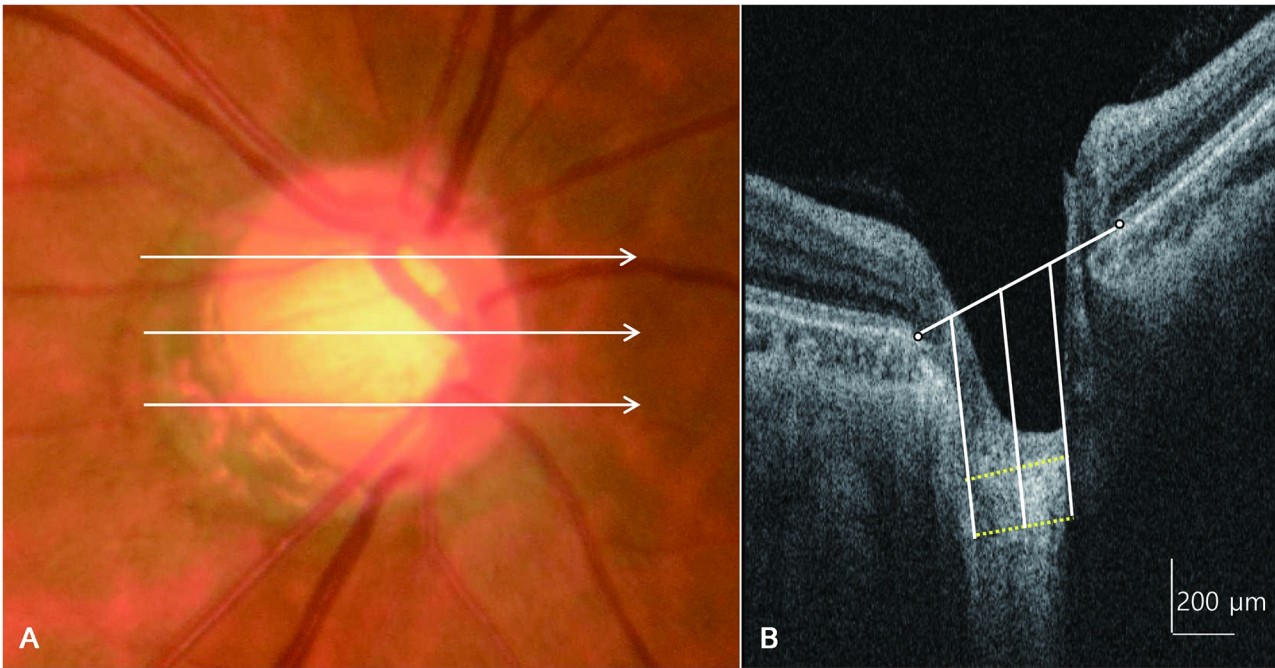

**Fig 1. A fundus photograph and horizontal cross-sectional B-scan images of the optic nerve head obtained using swept-source optical coherence tomography showing measurement of lamina cribrosa thickness.** The lamina cribrosa thickness is measured at the mid-superior, central, and mid-inferior levels of the optic nerve head (A). At each level, the thickness is measured at three points (center, 100 μm nasally, and 100 μm temporally), along a line perpendicular to a reference line connecting the two ends of Bruch's membrane (B).

LC. We also determined differences in LC evenness between BRVO and NTG and calculated the mean difference of ALCD from the mid-superior to mid-inferior portion of the central LC. A lower value indicated a more even, or flat, LC.

## OCTA of the peripapillary area

SS-OCTA was performed using the DRI-OCT triton (Topcon) to assess the structure of the peripapillary capillary plexus. Images were obtained using a 3 × 3-mm scan area centered on the optic disc. We analyzed the peripapillary capillary density in the superficial capillary plexus, which comprised the inner capillary layer of the retina; the deep capillary plexus, which comprised the outer capillary layer; as well as the choriocapillaris. The retinal vascular plexus described by Weinhaus et al. [18] was simplistically considered as two main layers, as described by Snodderly et al. [19] The inner capillary layer or the superficial capillary plexus started from the internal limiting membrane region and was of sufficient thickness to include the larger arteries, arterioles, capillaries, venules, and veins in the ganglion cell layer. It ended at the inner plexiform, which is the most superficial hyporeflective band. The outer capillary layer or the deep capillary plexus was identified by segmenting en-face images such that the inner boundary coincided with the inner nuclear layer and the outer boundary coincided with the midpoint of the outer plexiform layer [20, 21]. The choriocapillaris was identified by a 10-μm thick slab offset of 21 μm below the instrument generating the retinal pigment epithelial band (Fig 2) [22]. We used latest built-in software (IMAGEnet6) to generate OCT-angiograms which can provide improved detection sensitivity of low blood flow and reduced motion artifacts without compromising axial resolution [23]. Included OCTA images were subsequently

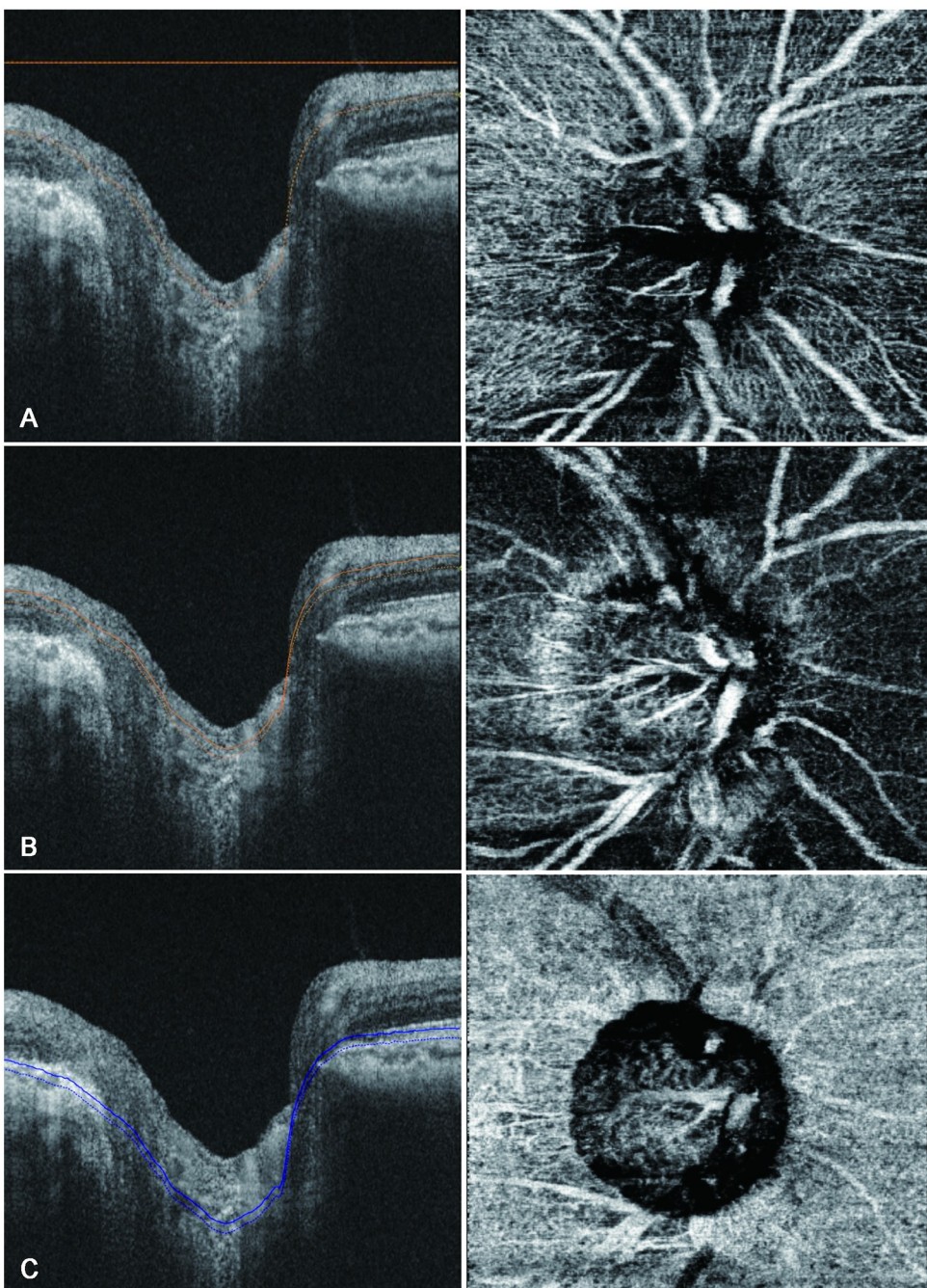

**Fig 2. B-scan images of the optic nerve (left) and peripapillary vessels (right) obtained using swept-source optical coherence angiography showing measurement of peripapillary vessel density.** The peripapillary vessel density is measured in the superficial capillary plexus, which comprises the inner capillary layer of the retina (A); the deep capillary plexus, which makes up the outer capillary layer (B); and the choriocapillaris (C).

exported in PNG format and imported into an i-solution image analysis program (iMT, i-Solution Inc).

Subsequently, each of the three layers was segmented into superotemporal, inferotemporal, superonasal, and inferonasal regions. Image processing and analysis consisted of 3 stages [24].

The first stage was extraction of the region of interest (ROI) around the optic nerve. The optic nerve was identified as a circle, and the outer margin was placed at 750 μm from the margin of this circle. In cases of myopia with a zone of peripapillary chorioretinal atrophy, the circle included the zone of atrophy, to minimize the influence on the peripapillary vessel density [25, 26]. The donut-shaped portion between the outer margin and the optic nerve margin was divided into four segments using Adobe Photoshop CS2 (Adobe Systems, Inc.), and finally, the donut-shaped ROI was extracted (Fig 3A and 3B). The second stage was detection and deletion of thick vessels from the ROI. We checked large vessels such as the inferior and superior nasal and the inferior and superior temporal retinal veins and arteries, and we excluded these vessels one-by-one using sub mode (deletion mode) with a built-in program in i-solution (Fig 3C and 3D). The third stage was estimation of the peripapillary vessel density, which is the ideal measure of capillaries per unit area. The desired image is the binary image, in which white pixels represent capillaries (without large vessels) and black pixels represent non-vessels. The peripapillary vessel density was calculated using the following formula: peripapillary vessel density (%) = $(N_w / A) \times 100$, where $N_w$ represents the number of white pixels and A represents the area of the selected image sector. As both the numerator and the denominator are pixel counts, the peripapillary vessel density is reported between 0% and 100% [27–30].

**Data analysis.** Within-visit repeatability of the peripapillary vessel density measurements was calculated using two sets of images that were sequentially obtained in a single visit. The interobserver reproducibility was calculated from measurements recorded for 10 randomly selected eyes by two independent observers (JBC and JMW) who were masked to the patients' information. Variability was assessed using the intraclass correlation coefficient (ICC) with a two-way random effects model.

The two groups were matched for age and the mean deviation prior to performing inter-group comparisons, for which independent t-tests were used. In analyzing peripapillary vessel density, we used one-way multivariate analysis of variance. All statistical analyses were performed using Statistical Package for the Social Sciences (SPSS) for Windows (version 21.0, SPSS, Inc., Chicago, IL, USA). A P-value of $< 0.05$ was considered statistically significant.

## Result

A total of 71 patients were initially included in this study. Of these, three were excluded from the BRVO group, since one patient showed the possibility of primary open-angle glaucoma concomitant with BRVO and the other two had poor-quality OCTA images. Four patients with poor-quality OCTA images were also excluded from the NTG group. Eventually, 21 eyes of 21 patients with BRVO and 43 eyes of 43 patients with NTG were analyzed. The age range of patients overall was 48–79 years (mean age: $64.81 \pm 8.81$ years). There were no significant differences in age, sex, and laterality between the two groups (P = 0.79, P = 0.42, and P = 0.30, respectively). Moreover, the two groups showed no significant differences in terms of a history of hypertension or diabetes mellitus and Humphrey visual field indices, such as the mean deviation and pattern standard deviation (Table 1).

### Comparison of LC properties

The ALCD at the mid-superior, central, and mid-inferior levels was significantly greater in the NTG group than in the BRVO group (P = 0.02, 0.01, and 0.01, respectively; Table 2).

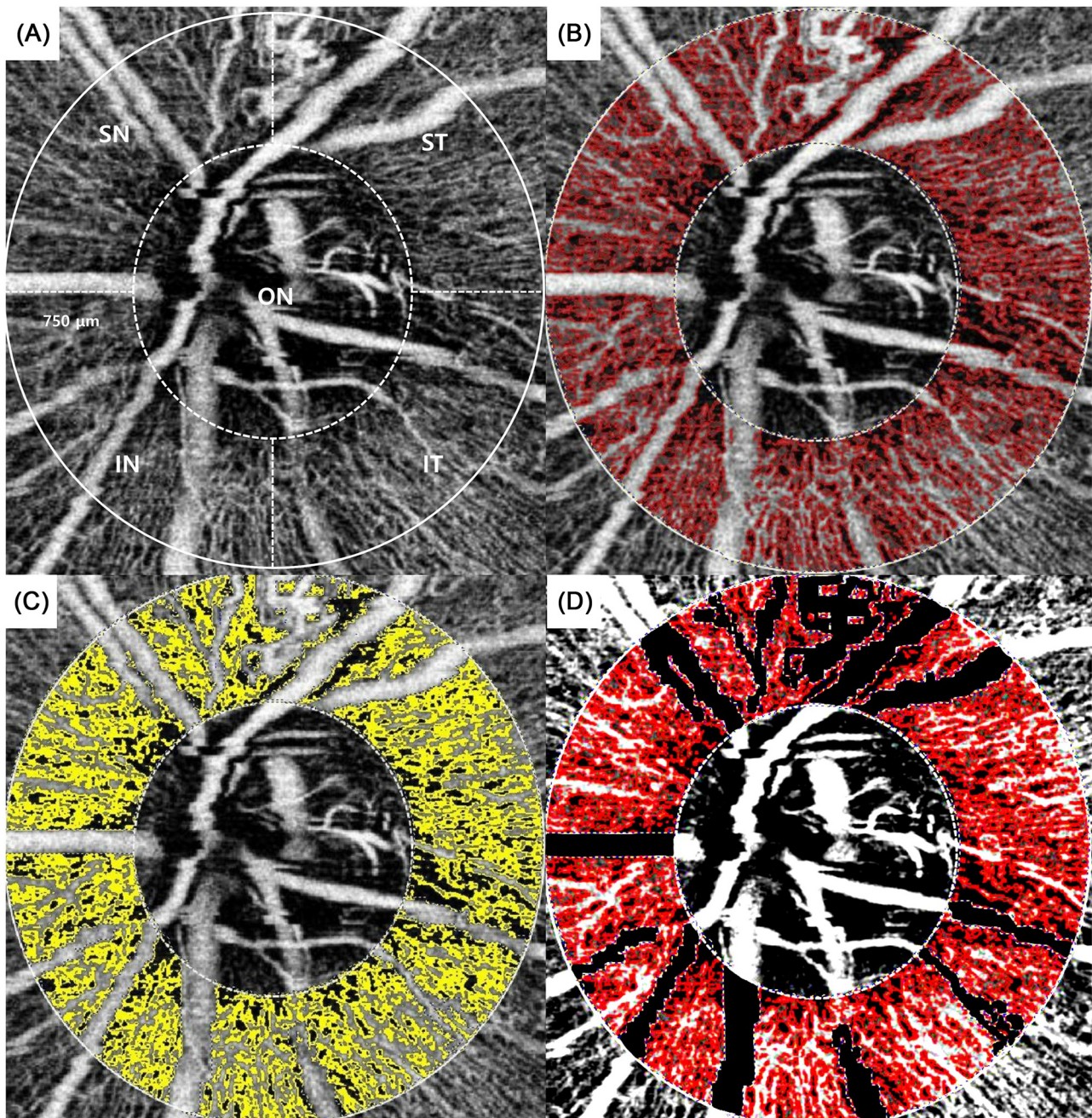

**Fig 3. Segmental valuation of the peripapillary vessel density.** The optic nerve was indicated as a circle, and the outer margin was set at 750 μm from this circle margin. The donut-shaped portion between the outer margin and the optic nerve margin was divided into four segments, namely the superotemporal, inferotemporal, superonasal, and inferonasal segments (A); each segment was extracted as a region of interest (ROI) (B); large vessels were detected and deleted (C); and finally, the binary ROI of the peripapillary vessels, with large vessels excluded, was retained (D). ON, optic nerve; ST, superotemporal; IT, inferotemporal; SN, superonasal; IN, inferonasal.

LCT at all levels was smaller in the NTG group than in the BRVO group, although the differences were not significant at any level (Table 3).

In the BRVO group, the superotemporal segment was the site of retinal vein occlusion in 16 patients (77%), the inferotemporal segment was affected in 4 (19%) and both the supero- and inferotemporal segments were affected in one patient (4%).

**Table 1. Baseline characteristics of BRVO and NTG patients.**

| | BRVO group (n = 21) | NTG group (n = 43) | *P* value |
|---|---|---|---|
| Age, year (mean ± SD) | 58.19 ± 10.75 | 57.39 ± 13.23 | 0.79 |
| Sex, men/women (% (n)) | 47 (10)/53 (11) | 58.1 (25)/41.9 (18) | 0.42 |
| Laterality (OD/OS) | 6/15 | 18/25 | 0.3 |
| S.E. (diopter) | -1.25± 0.85 | -2.45±1.12 | 0.3 |
| HTN (%) | 42.8 | 25.5 | 0.16 |
| DM (%) | 9.5 | 6.9 | 0.72 |
| HVF | | | |
| MD (dB) | -8.93 ± 5.17 | -8.61 ± 7.9 | 0.84 |
| PSD (dB) | 8.94 ± 2.68 | 7.60 ± 4.04 | 0.17 |

*$P < 0.05$.

BRVO: branch retinal vein occlusion, NTG, normal tension glaucoma; S.E.: Spherical equivalent, HTN: hypertension, DM: diabetic mellitus, HVF: Humphrey visual field, MD: mean deviation, PSD: pattern standard deviation, dB: decibel.

The mean difference in ALCD from the mid-superior to the mid-inferior levels of the central LC was significantly greater in the BRVO group than in the NTG group (74.8 ± 39.1 μm and 41.6 ± 57.1 μm, respectively; P = 0.03; Table 4).

## Comparisons of the peripapillary capillary density

The peripapillary capillary density in the superotemporal segment of the superficial capillary plexus was significantly lower in the BRVO group than in the NTG group (58.25 ± 14.24 and 68.90 ± 13.15, respectively; P = 0.005; Fig 4A). For all segments in the choriocapillaris, the vessel density was significantly lower in the NTG group than in the BRVO group (P < 0.05 for all; Fig 4). The vessel density in the inferotemporal segment of the superficial capillary plexus was lower in the NTG group than in the BRVO group, although the difference was not statistically significant (P = 0.401; Fig 4B). The NTG group generally tended to exhibit a lower vessel density in all segments in the deep layer than did the BRVO group; however, these differences were not statistically significant (Fig 4).

**Table 2. Comparing anterior lamina depth between BRVO and NTG patients.**

| | | BRVO group (n = 21) | NTG group (n = 43) | *P* value |
|---|---|---|---|---|
| Mid-superior | ALDn, μm | 329.4 ± 122.2 | 378.3 ± 141.6 | 0.23 |
| | ALDc, μm | 366.7 ± 119.9 | 450.9 ± 138.0 | 0.02* |
| | ALDt, μm | 223.5 ± 74.9 | 320.6 ± 123.1 | 0.02* |
| Central | ALDn, μm | 301.4 ± 106.2 | 345.7 ± 129.4 | 0.22 |
| | ALDc, μm | 322.2 ± 105.7 | 432.5 ± 134.0 | 0.01* |
| | ALDt, μm | 271.6 ± 107.6 | 309.5 ± 127.1 | 0.29 |
| Mid-inferior | ALDn, μm | 273.9 ± 93.8 | 301.2 ± 116.1 | 0.35 |
| | ALDc, μm | 292.9 ± 115.1 | 406.0 ± 123.0 | 0.01* |
| | ALDt, μm | 264.0 ± 100.3 | 298.5 ± 109.0 | 0.22 |

*$P < 0.05$.

BRVO: branch retinal vein occlusion, NTG: normal tension glaucoma, ALDn: anterior lamina depth of nasal side, ALDc: anterior lamina depth of central side, ALDt: anterior lamina depth length of temporal side.

**Table 3. Comparing lamina cribrosa thickness between BRVO and NTG patients.**

| | | BRVO group (n = 21) | NTG group (n = 43) | *P* value |
|---|---|---|---|---|
| Mid-superior | LCn, μm | 175.8 ± 75.6 | 169.2 ± 47.9 | 0.67 |
| | LCc, μm | 177.9 ± 78.8 | 165.5 ± 56.7 | 0.47 |
| | LCt, μm | 182.0 ± 56.28 | 157.9 ± 52.0 | 0.10 |
| Central | LCn, μm | 183.4 ± 63.3 | 173.2 ± 53.2 | 0.52 |
| | LCc, μm | 176.9 ± 57.4 | 152.7 ± 52.9 | 0.11 |
| | LCt, μm | 186.3 ± 81.8 | 170.7 ± 65.7 | 0.45 |
| Mid-inferior | LCn, μm | 176.8 ± 38.2 | 175.6 ± 53.1 | 0.91 |
| | LCc, μm | 178.5 ± 70.5 | 164.3 ± 65.3 | 0.44 |
| | LCt, μm | 172.9 ± 48.0 | 165.1 ± 62.5 | 0.58 |

*$P < 0.05$.

BRVO: branch retinal vein occlusion, NTG: normal tension glaucoma, LCn: lamina cribrosa thickness of nasal side, LCc: lamina cribrosa thickness of central side, LCt: lamina cribrosa thickness of temporal side.

## Repeatability and reproducibility

For peripapillary vessel density measurements, ICCs for within-visit repeatability and interobserver reproducibility with the i-solution® program were 0.956 and 0.975 (excellent), respectively.

## Discussion

In this study, we hypothesized that LC properties and peripapillary vessel density would differ between BRVO and NTG patients, and that this could help differentiate between the pathogeneses of the two conditions. Accordingly, we used SS-OCT and SS-OCTA to investigate this hypothesis; we indeed found that such differences exist (Fig 5).

ALCD was generally greater in the NTG than in the BRVO group, while LCT was insignificantly smaller in the former. Moreover, the LC was more irregular in BRVO, with its superior portion being the most common site of retinal vein occlusion. The optic nerve head is considered a biomechanically weak area because it coincides with discontinuity in the corneoscleral shell [31]. The LC is a mesh-like structure that is composed of a multilayered network of collagen fibers and occupies a hole in the sclera. It allows the nerve fibers of the optic nerve and retinal vessels to pass through the sclera and to exit the eye. Therefore, LCT can decrease in cases with ganglion cell degeneration and axonal damage with consequent inhibition of axonal transport [32, 33]. The mechanism involved is similar to that by which a column in a concrete building weakens upon removal of the reinforcing bars. The most typical disease associated with ganglion cell degeneration and axonal damage is glaucoma, particularly NTG. Several studies have reported that the LC and pre-LC are thinner in eyes with NTG than in healthy eyes [34–36]. Kim et al. [37] found that the anterior laminar insertion depth was greater than

**Table 4. Mean difference of anterior lamina depth from mid-superior to mid-inferior portion of central lamina cribrosa.**

| Mean difference | BRVO group (n = 21) | NTG group (n = 43) | *P* value |
|---|---|---|---|
| ALCD, μm | 74.8 ± 52.8 | 41.7 ± 46.7 | 0.02* |

*$P < 0.05$.

BRVO: branch retinal vein occlusion, NTG: normal tension glaucoma.

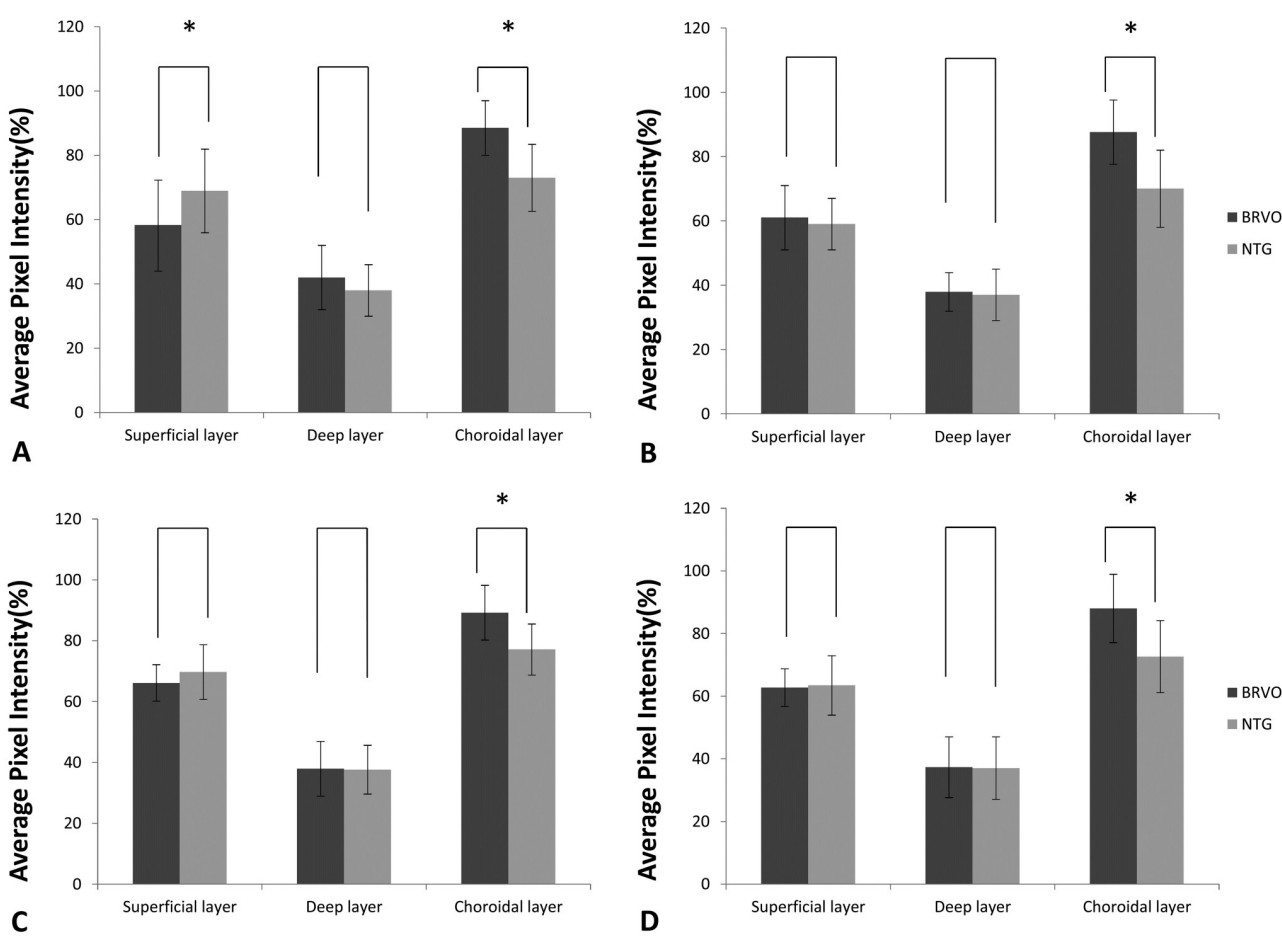

**Fig 4. Swept-source optical coherence angiography findings for the peripapillary vessel density in different layers in eyes with branch retinal vein occlusion (BRVO) and those with normal-tension glaucoma (NTG) (A, Superiotemporal; B, Inferotemporal; C, Superonasal; D Inferonasal area).** The vessel density in the superotemporal segment of the superficial capillary plexus is significantly lower in BRVO than in NTG (P = 0.008) (A). The vessel density in all segments in the choriocapillaris is significantly lower in NTG than in BRVO (P < 0.05) (A–D) (Error bar = standard deviation).

normal in eyes with NTG. The LC and pre-LC were also found to be thinner in eyes with BRVO, due to axonal damage and shallow optic disc excavation, than in healthy eyes [38]. However, in BRVO, which derives its name from the presence of occlusion in only a particular segment of the retinal vein, there is a tendency for the retinal nerve fiber layer to exhibit thinning only in the affected portion, while the unaffected portion of the same optic nerve does not show thinning [15]. A possible mechanism for this type of defect is that the retinal ischemic lesion may lead to focal axonal dysfunction and cause descending retinal ganglion cell degeneration and atrophy [39]. Moreover, BRVO predominantly occurs in the superotemporal segment [40], whereas the damage associated with glaucoma first involves the axonal bundles, with somewhat greater involvement of the inferior and superior poles of the optic disc and more diffuse and generalized damage than that observed in BRVO [41]. Consequently, we observed less LC thinning and disc cupping in our BRVO group than in our NTG group. Moreover, in eyes with BRVO, ALCD in the superior portion of the LC, which is the predominant site for retinal vein occlusion [40], is greater than that in the inferior portion. On the other hand, the LC shows overall uniformity in eyes with NTG (Fig 5E and 5J).

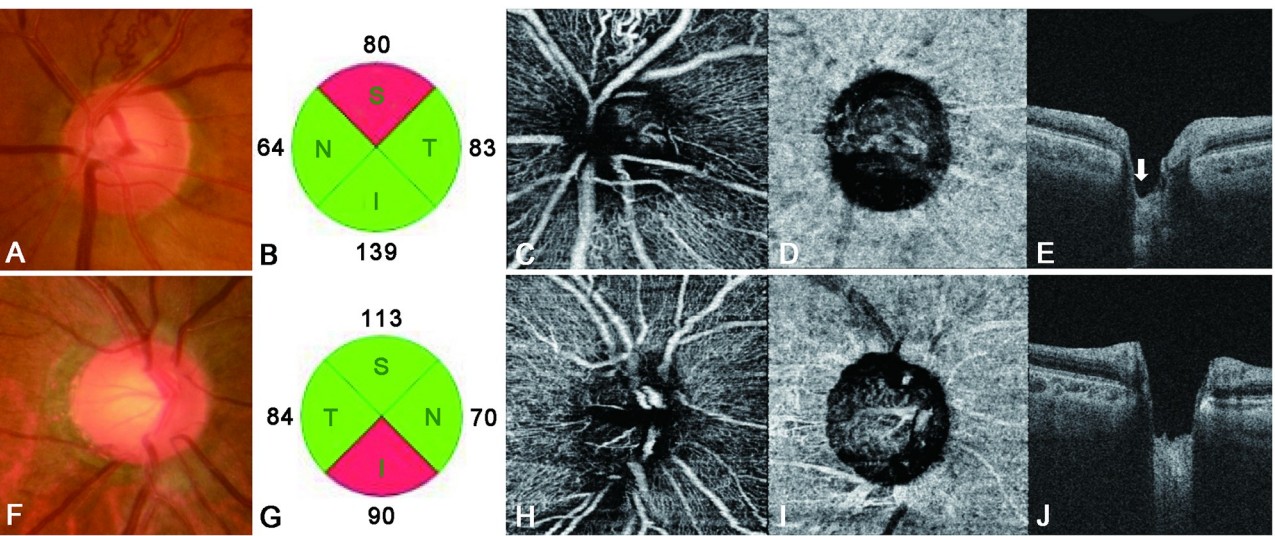

**Fig 5.** Lamina cribrosa properties and the peripapillary vessel density for representative cases of branch retinal vein occlusion (BRVO) and normal-tension glaucoma (NTG) (A–E). The left eye of a patient with BRVO, showing a mean deviation (Humphrey visual field [HVF]) of −5.21 dB (F–J). The right eye of a patient with NTG, showing a mean deviation (HVF) of −5.41 dB. The eye with BRVO shows superior disc notching (A) and a superior retinal nerve fiber layer (RNFL) defect (B). The peripapillary vessel density in the superotemporal segment of the superficial capillary plexus has decreased (C), with more superior disc cupping seen on a vertical B-scan (white arrow, E). The vessel density in the choriocapillaris is lower in the eye with NTG (I) than in the eye with BRVO. Moreover, the eye with NTG shows greater disc cupping and a more uniform LC (J) than does the eye with BRVO.

OCTA has inherent advantages due to its capacity for optical dissection and visualization of the blood flow in various layers of the retina [20]. Using OCTA, we have demonstrated a difference in the peripapillary vessel density between NTG and BRVO patients and showed that, in the BRVO group, the vessel density in the superotemporal segment (where vessel occlusion was more frequent [40]) of the superficial capillary plexus was significantly lower than that in the NTG group (Fig 4A). This could be associated with a decrease in the peripapillary retinal microvasculature at the site of the retinal nerve fiber layer defect [13, 42]. The vessel density in the inferotemporal segment (which is more frequently affected in NTG patients) of the superficial capillary plexus was lower in NTG than in BRVO patients, although the difference was not significant. In the choriocapillaris, the peripapillary vessel density in all segments was significantly lower in NTG than in BRVO patients (Figs 4, 5D and 5I), probably because of the unique blood supply in the optic nerve head. The surface nerve fiber layer is primarily supplied by retinal arterioles, and the ganglion cell layer is primarily supplied by the inner vascular plexus of the retina [43]. The choroidal layer of the peripapillary vessels supplies the prelaminar region; some of these vessels form the arterial circle of Zinn–Haller and supplies the LC region [44, 45]. Therefore, in BRVO, secondary damage to ganglion cells and the retinal nerve fiber layer may occur around the primary injured vessel, with a consequent decrease in peripapillary capillaries. On the other hand, the LC is thought to be the primary affected site in NTG. Accordingly, we speculate that, not only the density of the peripapillary capillaries (which supply consequently damaged ganglion cells and retinal nerve fibers), but also that associated with the LC, is decreased in NTG. The decrease in superficial peripapillary capillaries in BRVO may be solely due to damage to the ganglion cells and retinal nerve fiber layer. However, the decrease in the choriocapillaris vessel density in NTG may not be associated with this type of damage; rather, it may be associated with the etiology of NTG. Both BRVO and NTG present similar clinical pictures, with ganglion cell and nerve fiber layer thinning and disc atrophy

[46]; yet the blood supply to the LC is significantly decreased only in NTG. This insufficient blood supply and vascular dysregulation may induce damage to the retinal nerve fibers, which pass through the LC, with consequent damage to secondary ganglion cells. Accordingly, we assume that NTG is directly caused by a decrease in the choriocapillaris vessel density, which is consistent with several studies that have indirectly implicated a low perfusion pressure and vascular dysregulation as important events in the pathogenesis of NTG [47–49].

This study hads some limitations. First, blood flow quantification was not possible with our OCTA device, and we were unable to determine the precise blood flow in the peripapillary vessels. Second, our OCT device does not provide software for automatic calculations of vessel density. However, we used i-solution® software, which has been successfully used for similar calculations in other fields; moreover, the interobserver reproducibility obtained using this program was excellent. Third, the most commonly affected sites differed between the two conditions. Therefore, to increase understanding of differences in the pathogenesis of BRVO and NTG, the differences between these two conditions in the same pathological location, such as the inferotemporal or superotemporal segment, should be analyzed further. Finally, although choroidal vessels comprise the choriocapillaris, Sattler's layer, and Haller's layer, SS-OCTA cannot clearly depict whole choroidal vessels, unlike laser speckle flowmetry [50]. We therefore considered the choriocapillaris as representing choroidal blood flow as a whole.

Another limitation may be the different number of patients between the two groups.

In conclusion, we demonstrated that BRVO and NTG have different LC properties and peripapillary vessel densities. LC cupping is more pronounced in NTG than in BRVO, and focal cupping is more common in BRVO than in NTG, due to the focal damage to retinal nerve fibers passing through the LC in BRVO. BRVO commonly exhibits decreased capillaries in the superotemporal segment of the superficial capillary plexus, whereas NTG commonly exhibits decreased vessels in the choriocapillaris. Finally, deformation and thinning of the LC can be due to damaged and lost axonal tissue and inadequate blood supply to the LC, caused by glaucomatous damage, particularly in NTG. Further study of diseases that are clinically similar to NTG could enhance our understanding of the pathogenesis of glaucoma, particularly NTG.

## Author Contributions

**Conceptualization:** Chang Kyu Lee.

**Data curation:** Je Moon Woo, Jae Bong Cha, Chang Kyu Lee.

**Formal analysis:** Chang Kyu Lee.

**Funding acquisition:** Chang Kyu Lee.

**Investigation:** Je Moon Woo, Jae Bong Cha, Chang Kyu Lee.

**Methodology:** Je Moon Woo, Jae Bong Cha, Chang Kyu Lee.

**Project administration:** Chang Kyu Lee.

**Resources:** Chang Kyu Lee.

**Software:** Chang Kyu Lee.

**Supervision:** Chang Kyu Lee.

**Validation:** Chang Kyu Lee.

**Visualization:** Chang Kyu Lee.

**Writing – original draft:** Chang Kyu Lee.

**Writing – review & editing:** Chang Kyu Lee.

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
