## [Decision Letter · Decision Letter 0]

3 Apr 2020

PONE-D-20-01336

Comparison of lamina cribrosa properties and the peripapillary vessel density between branch retinal vein occlusion and normal-tension glaucoma

PLOS ONE

Dear Dr. Lee,

Thank you for submitting your manuscript to PLOS ONE. After careful consideration, we feel that it has merit but does not fully meet PLOS ONE’s publication criteria as it currently stands. Therefore, we invite you to submit a revised version of the manuscript that addresses the points raised during the review process.

Both the reviewers have recommended major changes that need to be incorporated in a revised manuscript. 

We would appreciate receiving your revised manuscript by May 18 2020 11:59PM. To enhance the reproducibility of your results, we recommend that if applicable you deposit your laboratory protocols in protocols.io, where a protocol can be assigned its own identifier (DOI) such that it can be cited independently in the future. For instructions see: http://journals.plos.org/plosone/s/submission-guidelines#loc-laboratory-protocols

We look forward to receiving your revised manuscript.

Kind regards,

Sanjoy Bhattacharya

Academic Editor

PLOS ONE

Journal Requirements:

2. In ethics statement in the manuscript and in the online submission form, please provide additional information about the patient records/samples used in your retrospective study. Specifically, please ensure that you have discussed whether all data/samples were fully anonymized before you accessed them and/or whether the IRB or ethics committee waived the requirement for informed consent. If patients provided informed written consent to have data/samples from their medical records used in research, please include this information.

3. Thank you for including your ethics statement:

This retrospective cohort study adhered to the tenets of the Declaration of Helsinki and was approved by the Institutional Review Board (UUH 2017-09-037).

5. Your ethics statement must appear in the Methods section of your manuscript. If your ethics statement is written in any section besides the Methods, please move it to the Methods section and delete it from any other section. Please also ensure that your ethics statement is included in your manuscript, as the ethics section of your online submission will not be published alongside your manuscript.

Reviewers' comments:

Reviewer's Responses to Questions

**Comments to the Author**

1. Is the manuscript technically sound, and do the data support the conclusions?

Reviewer #1: Yes

Reviewer #2: Partly

2. Has the statistical analysis been performed appropriately and rigorously? 

Reviewer #1: Yes

Reviewer #2: I Don't Know

3. Have the authors made all data underlying the findings in their manuscript fully available?

Reviewer #1: Yes

Reviewer #2: No

4. Is the manuscript presented in an intelligible fashion and written in standard English?

Reviewer #1: Yes

Reviewer #2: Yes

5. Review Comments to the Author

Reviewer #1: Lines 43-44: Globally the prevalence of AMD is higher than RVO, please correct and cite.

Lines 48 – 50. Authors should be careful in associating pathophysiology of BRVO with NTG. These two diseases could maybe be associated with similar signs evaluated with a routine fundus ophthalmoscopy, but the mechanism can be different. Authors should clarify that NTG is more often associated with status of hypotension, headache and generalized vasoconstriction of blood vessels especially on the extremities.

Lines 89-91: Authors should mention OCT tomography as useful tool to support the diagnosis of BRVO detecting retinal thinning and PAMM lesions.

(En Face Optical Coherence Tomography Analysis to Assess the Spectrum of Perivenular Ischemia and Paracentral Acute Middle Maculopathy in Retinal Vein Occlusion.

Ghasemi Falavarjani K, Phasukkijwatana N, Freund KB, Cunningham ET Jr, Kalevar A, McDonald HR, Dolz-Marco R, Roberts PK, Tsui I, Rosen R, Jampol LM, Sadda SR, Sarraf D.

Am J Ophthalmol. 2017 May;177:131-138. doi: 10.1016/j.ajo.2017.02.015. Epub 2017 Feb 22)

Lines 113-114: Can you specify the manufacturer’s scale given it is different among different companies?

Lines 156-158: Authors should specify the thickness of the slab they used to segment the choriocapillaris.

Lines 200-201: Limitation of this retrospective study is the different size of the two groups.

I would add a paragraph describing the details of the algorithm used to perform the analyses.

Was the peripapillary atrophy taken into consideration? Was it excluded from the analyses? Differences can also be affected by different extension of the peripapillary atrophy in the two groups so I would repeat the analyses and correct for that factor.

Reviewer #2: Major Concern:

Method of estimating capillary density

In this study capillary density is being measured using an average pixel intensity, which is a very crude measurement and could be affected by many confounding factors (for example, image contrast and levels due to slightly different quality of images could have a significant effect). How does exclusion of large vessels (briefly noted in lines 182:183) affect these measurements? Simple removal of large vessels (replacing those pixels with black) would have a significant impact on the average pixel intensity. Moreover, based on the description of methods in lines 167:170, the area of retina being analyzed will be different for each subject based on different Optic Nerve (ON) circle sizes. How was this controlled for? Is there a difference in ON size between NTG group and the BRVO group? In a normal eye does the density of peripapillary capillaries/choriocapillaris changes with distance from ON?

None the sources cited (references 23-36) validate this method for estimation of capillary density, let alone retinal capillary density with OCT. There are numerous Ophthalmology OCT-A studies that look at capillary density that have developed more accurate methods for measuring capillary density, and one of these validated techniques should be used in this paper. For examples of established methods see:

Geyman LS, Garg RA, Suwan Y, et al. Peripapillary perfused capillary density in primary open-angle glaucoma across disease stage: an optical coherence tomography angiography study. British Journal of Ophthalmology. 2017;101:1261-1268.

Fard et al. (2018) Pattern of peripapillary capillary density loss in ischemic optic neuropathy compared to that in primary open-angle glaucoma. PLoS ONE 13(1): e0189237. https://doi.org/10.1371/journal.pone.0189237

Mansoori et al. Radial Peripapillary Capillary Density Measurement Using Optical Coherence Tomography Angiography in Early Glaucoma. J of Glaucoma. 2017; 26(5):438-443. doi: 10.1097/IJG.0000000000000649

Mansoori et al. Measurement of Radial Peripapillary Capillary Density in the Normal Human Retina Using Optical Coherence Tomography Angiography. J Glaucoma. 2017 Mar;26(3):241-246. doi: 10.1097/IJG.0000000000000594.

Florence Coscas, Alexandre Sellam, Agnès Glacet-Bernard, Camille Jung, Mathilde Goudot, Alexandra Miere, Eric H. Souied; Normative Data for Vascular Density in Superficial and Deep Capillary Plexuses of Healthy Adults Assessed by Optical Coherence Tomography Angiography. Invest. Ophthalmol. Vis. Sci. 2016;57(9):OCT211-OCT223. doi: https://doi.org/10.1167/iovs.15-18793.

Jorge S. Andrade Romo, Rachel E. Linderman, Alexander Pinhas, Joseph Carroll, Richard B. Rosen, Toco Y. P. Chui; Novel Development of Parafoveal Capillary Density Deviation Mapping using an Age-Group and Eccentricity Matched Normative OCT Angiography Database. Trans. Vis. Sci. Tech. 2019;8(3):1. doi: https://doi.org/10.1167/tvst.8.3.1.

Other comments:

Line 48-50:

The statement “This suggests that BRVO and glaucoma, particularly normal-tension glaucoma (NTG), share a common pathological mechanism” is not supported by the provided citations.

The cited studies show an association between elevated IOP/increase CD ratio and retinal vein occlusion (RVO), but do not imply that BRVO and glaucoma have same pathological mechanism. They suggests an increased risk of BRVO with glaucoma caused by changes in the retina that occur due to the glaucoma or directly due to elevated IOP (cause and effect, rather than a common underlying etiology). Furthermore, these studies suggest a link between RVO and elevated IOP, not normal tension glaucoma.

If you are indeed suggesting that BRVO and NTG share a common pathological mechanism please provide more evidence. Also please provide a more in depth explanation as to what the common underlying mechanism is, and why you are focused on NTG rather than glaucoma with elevated IOP

Line 67:

Suggest avoiding using “accordingly” to start two paragraphs in a row; It can simply be deleted here.

Methods Questions:

For the 43 patients with NTG, which eye was selected and why? Please elaborate.

Line 105-106:

This statement is poorly worded, and seems to state that the depth the OCT can scan is only 8um in depth, which is clearly not true. It is the In-depth resolution of the device that is 8um. Please clarify.

Line 107:

Suggest changing this from “constant signal strength of the posterior pole” to constant signal strength throughout the posterior pole”

Line 153:

“Hypoechoic lesion” is the wrong term. Hypo/hyperechoic refer to ultrasound, not OCT which uses light rather than sound. Please use the correct terms (hyper/hyporeflective). Additionally, the inner plexiform layer is not a lesion, it should be referred to as a band. Please correct

TABLE 1:

No need to include both % and (n) for sex, as one can easily be calculated from the other. Recommend consistent reporting of data, use the same format that you choose for reporting OD/OS.

Line 218-219:

Please include the units of ALCD mean difference

Line 217 and Table 4:

ALCD vs ALDc Do these acronyms represent the same thing? If so, why use two acronyms? Please clarify the difference. Also suggest using a different acronym to distinguish mean ALDc from ALDc used in Table 2

Line 268-270:

You state “… LTC can decrease in cases with ganglion cell degeneration and axonal damage, with consequent inhibition of axonal transport.”

Are you saying inhibition of axonal transport a consequence of RGC degeneration or is it a cause? This is a controversial topic and this statement should be re-worded or supported by citations.

Line 270-271

The analogy comparing LC weakening to a concrete building is very clever. I have never heard it before, but it is great way to visualize this.

Line 281-282

An ischemic lesion would not cause “focal axonal transection,” which implies cutting. Please reword this.

6. PLOS authors have the option to publish the peer review history of their article (what does this mean?). If published, this will include your full peer review and any attached files.

Reviewer #1: Yes: Giulia Corradetti

Reviewer #2: No

---

## [Author Response · Author response to Decision Letter 0]

4 Jun 2020

Dear reviewers & editor 

First of all, we would like to thank the reviewers for their time and valuable comments on our manuscript. We have addressed our opinions on each comment of the two reviewers in this response letter, and made several changes and corrections to our original manuscript. 

We tried to revise our manuscript according to reviewers’ suggestions as much as possible, and we hope that the revisions in the manuscript and our accompanying responses will be sufficient to make our manuscript suitable for publication in PLoS one. 

We shall look forward to hearing from you at your earliest convenience.

Journal Requirements:

: We have edited our manuscript to meet PLOS ONE’s style requirements as the PLOS ONE style templates

2. In ethics statement in the manuscript and in the online submission form, please provide additional information about the patient records/samples used in your retrospective study. Specifically, please ensure that you have discussed whether all data/samples were fully anonymized before you accessed them and/or whether the IRB or ethics committee waived the requirement for informed consent. If patients provided informed written consent to have data/samples from their medical records used in research, please include this information.

: Thank you for your comment. All data were fully anonymized before we accessed them and this study was retrospective study therefore, our IRB committee waived the requirement for informed consent. And we added this sentence at method part in manuscript. 

3. Thank you for including your ethics statement:

This retrospective cohort study adhered to the tenets of the Declaration of Helsinki and was approved by the Institutional Review Board (UUH 2017-09-037).

: We amended our current ethics statement (UUH 2017-09-037-001) and included full name of the ethics committee to revised manuscript. 

: We add the same text to the ethics statement filed of the submission form 

: Even though this study was retrospective study and all data were anonymized before we accessed them, all data in this study was human research participant data and we did not take informed consent from patients to provide journal their data. Therefore, it is hard to update our data. 

5. Your ethics statement must appear in the Methods section of your manuscript. If your ethics statement is written in any section besides the Methods, please move it to the Methods section and delete it from any other section. Please also ensure that your ethics statement is included in your manuscript, as the ethics section of your online submission will not be published alongside your manuscript.

: Our ethics statement was appeared in methods section as “This retrospective cohort study adhered to the tenets of the Declaration of Helsinki and was approved by the ulsan university hospital Institutional Review Board (UUH 2017-09-037-001). All subjects were enrolled between June 2017 and September 2018.” 

Reviewers' comments:

Reviewer's Responses to Questions

 Review Comments to the Author

Reviewer #1: Lines 43-44: Globally the prevalence of AMD is higher than RVO, please correct and cite.

: First of all, thank you for your precious time on reviewing our manuscript. As you mentioned, we corrected that sentence and we added right citation.

Lines 48 – 50. Authors should be careful in associating pathophysiology of BRVO with NTG. These two diseases could maybe be associated with similar signs evaluated with a routine fundus ophthalmoscopy, but the mechanism can be different. Authors should clarify that NTG is more often associated with status of hypotension, headache and generalized vasoconstriction of blood vessels especially on the extremities.

: Thanks for your kind and detail review. As you mentioned, this sentence could lead to misunderstanding to reader, therefore, we changed this sentence more clearly and specifically and we added above sentence which you recommended.“This suggests that BRVO and glaucoma, particularly normal-tension glaucoma (NTG), may have a similar pathological mechanism such as blood flow insufficiency, NTG is more often associated with status of hypotension, headache and generalized vasoconstriction of blood vessels especially on the extremities though”

Lines 89-91: Authors should mention OCT tomography as useful tool to support the diagnosis of BRVO detecting retinal thinning and PAMM lesions.

(En Face Optical Coherence Tomography Analysis to Assess the Spectrum of Perivenular Ischemia and Paracentral Acute Middle Maculopathy in Retinal Vein Occlusion.

Ghasemi Falavarjani K, Phasukkijwatana N, Freund KB, Cunningham ET Jr, Kalevar A, McDonald HR, Dolz-Marco R, Roberts PK, Tsui I, Rosen R, Jampol LM, Sadda SR, Sarraf D.

Am J Ophthalmol. 2017 May;177:131-138. doi: 10.1016/j.ajo.2017.02.015. Epub 2017 Feb 22)

: As you mentioned, we added OCT tomography as tool to support the diagnosis of BRVO and also add citation what you recommended. 

Lines 113-114: Can you specify the manufacturer’s scale given it is different among different companies?

:Manufacturer have recommended that image quality value must be 30 or higher, And we wanted to get high quality image, So we selected images which’s image value were over 35 

Lines 156-158: Authors should specify the thickness of the slab they used to segment the choriocapillaris.

:As you mentioned, we specified slab of choriocapillary segement as “ Slab of choriocapillaris was identified by imaging a 23-μm area posterior from the Bruch membrane by automated segmentation “

Lines 200-201: Limitation of this retrospective study is the different size of the two groups.

: We added this sentence at limitation of our manuscript as you mentioned 

I would add a paragraph describing the details of the algorithm used to perform the analyses.

: We used similar analyzing method which previous researcher used with DRI OCT Triton, so we think that we fully described how to analyze the data. However, I added some detail contents required for analysis with DRI OCT in method section 

Reference) 1. Tang FY et al. Determinants of quantitative optical coherence tomography angiography metrics in patients with diabetes. Sci Rep.2017 Mya 31;7(1) 

2. Sun Z et al. OCT angiography metrics predict progression of diabetic retinopathy and development of diabetic macular edema: a prospective study. Ophthalmology.2019 Dec ; 126 

Was the peripapillary atrophy taken into consideration? Was it excluded from the analyses? Differences can also be affected by different extension of the peripapillary atrophy in the two groups so I would repeat the analyses and correct for that factor.

: We already considered peripapillary atrophy (PPA), especially beta- zone before analyzing of this study. Therefore, we excluded high myopia to avoid including PPA cases, because PPA is frequently related with myopia, especially high myopia. Fortunately, mean of spherical equivalent of both group(BRVO, NTG) were not severe ( -1.25 diopter, -2.45 diopter respectively). Moreover, in case of PPA presentation, we manually controlled circle for optic nerve circumference to include PPA also, not to analyze PPA to peripapillary capillaries before we analyzed data. 

Reviewer #2: Major Concern:

Method of estimating capillary density

In this study capillary density is being measured using an average pixel intensity, which is a very crude measurement and could be affected by many confounding factors (for example, image contrast and levels due to slightly different quality of images could have a significant effect). How does exclusion of large vessels (briefly noted in lines 182:183) affect these measurements? Simple removal of large vessels (replacing those pixels with black) would have a significant impact on the average pixel intensity. Moreover, based on the description of methods in lines 167:170, the area of retina being analyzed will be different for each subject based on different Optic Nerve (ON) circle sizes. How was this controlled for? Is there a difference in ON size between NTG group and the BRVO group? In a normal eye does the density of peripapillary capillaries/choriocapillaris changes with distance from ON?

None the sources cited (references 23-36) validate this method for estimation of capillary density, let alone retinal capillary density with OCT. There are numerous Ophthalmology OCT-A studies that look at capillary density that have developed more accurate methods for measuring capillary density, and one of these validated techniques should be used in this paper. For examples of established methods see:

Geyman LS, Garg RA, Suwan Y, et al. Peripapillary perfused capillary density in primary open-angle glaucoma across disease stage: an optical coherence tomography angiography study. British Journal of Ophthalmology. 2017;101:1261-1268.

Fard et al. (2018) Pattern of peripapillary capillary density loss in ischemic optic neuropathy compared to that in primary open-angle glaucoma. PLoS ONE 13(1): e0189237. https://doi.org/10.1371/journal.pone.0189237

Mansoori et al. Radial Peripapillary Capillary Density Measurement Using Optical Coherence Tomography Angiography in Early Glaucoma. J of Glaucoma. 2017; 26(5):438-443. doi: 10.1097/IJG.0000000000000649

Mansoori et al. Measurement of Radial Peripapillary Capillary Density in the Normal Human Retina Using Optical Coherence Tomography Angiography. J Glaucoma. 2017 Mar;26(3):241-246. doi: 10.1097/IJG.0000000000000594.

Florence Coscas, Alexandre Sellam, Agnès Glacet-Bernard, Camille Jung, Mathilde Goudot, Alexandra Miere, Eric H. Souied; Normative Data for Vascular Density in Superficial and Deep Capillary Plexuses of Healthy Adults Assessed by Optical Coherence Tomography Angiography. Invest. Ophthalmol. Vis. Sci. 2016;57(9):OCT211-OCT223. doi: https://doi.org/10.1167/iovs.15-18793.

Jorge S. Andrade Romo, Rachel E. Linderman, Alexander Pinhas, Joseph Carroll, Richard B. Rosen, Toco Y. P. Chui; Novel Development of Parafoveal Capillary Density Deviation Mapping using an Age-Group and Eccentricity Matched Normative OCT Angiography Database. Trans. Vis. Sci. Tech. 2019;8(3):1. doi: https://doi.org/10.1167/tvst.8.3.1.

: Thanks for your valuable critiques and providing novel citation. However, in citation what you showed, they used different SS-OCT, Avanti OCT. It meant that built in program to show image was totally different with DRI –OCT. Moreover, image analyzing program such as SSAD, MATLAB could not be used to our study. Therefore, we looked for best way which extracted image and anazlyzed image with minimally compromising data. So, we took idea from quotation which we citied in manuscript. And our method is similar with tang ‘s way 1,2, because they used DRI-OCT instead using Avanti-OCT. For example, we used the latest built-in software (IMAGEnet6) to generated OCT-angiograms which can provide improved detection sensitivity of low blood flow and reduced motion artifacts without compromising axial resolution.3 Moreover, we tried to do our best to reduced confounding factors which can compromise data. For example, In cases of PPA, we excluded high myopia to avoid including PPA cases, because PPA is frequently related with myopia, especially high myopia. Fortunately, mean of spherical equivalent of both group(BRVO, NTG) were not severe ( -1.25 diopter, -2.45 diopter respectively). Moreover, in case of PPA presentation, we manually controlled circle for optic nerve circumference to include PPA also, not to analyze PPA to peripapillary capillaries before we analyzed data. With same context, there were no significant difference of ON size between both group. And when we excluded large vessels manually with i-solution program, there was no changing with pixel average. Also we did check intraclass correlation coefficient and Within-visit repeatability for reducing data compromising. Finally, before analyzing data with i-solution, we already noticed difference of OCTA between both group by checking manually, so it was nature and clear that same results was showed after analyzing with analyzing program, because analyzing program was one of artificial program just to detect factors which was shown in real life. 

Reference) 1. Tang FY et al. Determinants of quantitative optical coherence tomography angiography metrics in patients with diabetes. Sci Rep.2017 Mya 31;7(1) 

2. Sun Z et al. OCT angiography metrics predict progression of diabetic retinopathy and development of diabetic macular edema: a prospective study. Ophthalmology.2019 Dec ; 126 

3. Stanga, P.E et al. Swept-source optical coherence tomography angio(Topcon Corp.Japan): Technology Review. Dev ophthalmol 56,13-17(2016) 

Other comments:

Line 48-50:

The statement “This suggests that BRVO and glaucoma, particularly normal-tension glaucoma (NTG), share a common pathological mechanism” is not supported by the provided citations.

The cited studies show an association between elevated IOP/increase CD ratio and retinal vein occlusion (RVO), but do not imply that BRVO and glaucoma have same pathological mechanism. They suggests an increased risk of BRVO with glaucoma caused by changes in the retina that occur due to the glaucoma or directly due to elevated IOP (cause and effect, rather than a common underlying etiology). Furthermore, these studies suggest a link between RVO and elevated IOP, not normal tension glaucoma.

If you are indeed suggesting that BRVO and NTG share a common pathological mechanism please provide more evidence. Also please provide a more in depth explanation as to what the common underlying mechanism is, and why you are focused on NTG rather than glaucoma with elevated IOP

: Thanks for your kind and detail review. As you mentioned , this sentence could lead to misunderstanding to reader, therefore, we changed this sentence more clearly and specifically and we we added above sentence which you recommended.“ This suggests that BRVO and glaucoma, particularly normal-tension glaucoma (NTG), have a similar pathological mechanism such as blood flow insufficiency, NTG is more often associated with status of hypotension, headache and generalized vasoconstriction of blood vessels especially on the extremities though “

Line 67:

Suggest avoiding using “accordingly” to start two paragraphs in a row; It can simply be deleted here.

: As you mentioned, we deleted “accordingly” 

Methods Questions:

For the 43 patients with NTG, which eye was selected and why? Please elaborate.

: We selected eye of NTG by following criteria 1) eye of unilateral NTG patients 2) In case of bilateral NTG patients, we selected eye with higher resolution OCT image 3) In case of similar resolution of OCT image with bilateral NTG patients, we selected right eye to minimalize selection bias. We add this sentence to method part in our manuscript.

Line 105-106:

This statement is poorly worded, and seems to state that the depth the OCT can scan is only 8um in depth, which is clearly not true. It is the In-depth resolution of the device that is 8um. Please clarify.

: As you mentioned , we changed as a axial resolution of 8 μm

Line 107:

Suggest changing this from “constant signal strength of the posterior pole” to constant signal strength throughout the posterior pole”

: As you mentioned , we changed to “constant signal strength throughout the posterior pole”

Line 153:

“Hypoechoic lesion” is the wrong term. Hypo/hyperechoic refer to ultrasound, not OCT which uses light rather than sound. Please use the correct terms (hyper/hyporeflective). Additionally, the inner plexiform layer is not a lesion, it should be referred to as a band. Please correct

: We totally agree with your recommendation, so, we corrected from hypoechoic to hyporeflective and we deleted layer from inner plexiform layer.

TABLE 1:

No need to include both % and (n) for sex, as one can easily be calculated from the other. Recommend consistent reporting of data, use the same format that you choose for reporting OD/OS.

: As you recommend, we deleted % of sex to show same format that we chose for reporting OD/OS. 

Line 218-219:

Please include the units of ALCD mean difference

: As you recommend, we include the units of ALCD mean difference as μm

Line 217 and Table 4:

ALCD vs ALDc Do these acronyms represent the same thing? If so, why use two acronyms? Please clarify the difference. Also suggest using a different acronym to distinguish mean ALDc from ALDc used in Table 2

: AS you commended , this terms could be confused . ALCD was anterior laminal cribrosa depth and mean difference of ALCD was calculated values of mid-superior level of anterior lamina depth of central side (ALDc) minus values of mid-inferior level of anterior lamina depth of central side(ALDc). So, we wanted to emphasize that we used anterior lamina depth of central side (ALDc). However, as you mentioned, it may be confused, so, we changed ALDc in table to ALCD because using ALDc was described at title of Table. 

Line 268-270:

You state “… LTC can decrease in cases with ganglion cell degeneration and axonal damage, with consequent inhibition of axonal transport.”

Are you saying inhibition of axonal transport a consequence of RGC degeneration or is it a cause? This is a controversial topic and this statement should be re-worded or supported by citations.

: Quigley et al. said that the actual cause of early optic nerve head cupping in glaucoma appears to be the loss of axonal tissue in their studies. So, we added this studies as citations. 

Line 270-271

The analogy comparing LC weakening to a concrete building is very clever. I have never heard it before, but it is great way to visualize this.

: Thanks for your comment

Line 281-282

An ischemic lesion would not cause “focal axonal transection,” which implies cutting. Please reword this.

 : In Alshareef’s study( reference 37) ,they also expressed similar sentence , however it may cause misunderstanding , so we changed transection to dysfunction 

 Thank you for your valuable critiques and positive comments on our manuscript. We hope that the revisions made according to your comments may fulfill the requirements for publication in PLOS one.

---

## [Decision Letter · Decision Letter 1]

30 Jun 2020

PONE-D-20-01336R1

Comparison of lamina cribrosa properties and the peripapillary vessel density between branch retinal vein occlusion and normal-tension glaucoma

PLOS ONE

Dear Dr. Lee,

Thank you for submitting your manuscript to PLOS ONE. After careful consideration, we feel that it has merit but does not fully meet PLOS ONE’s publication criteria as it currently stands. Therefore, we invite you to submit a revised version of the manuscript that addresses the points raised during the review process.

Reviewer#2 have raised a number of critical issues that need to be addressed satisfactorily. In addition, Reviewer#1 has raised issues that also need to be addressed. 

We look forward to receiving your revised manuscript.

Kind regards,

Sanjoy Bhattacharya

Academic Editor

PLOS ONE

Reviewers' comments:

Reviewer's Responses to Questions

**Comments to the Author**

1. If the authors have adequately addressed your comments raised in a previous round of review and you feel that this manuscript is now acceptable for publication, you may indicate that here to bypass the “Comments to the Author” section, enter your conflict of interest statement in the “Confidential to Editor” section, and submit your "Accept" recommendation.

Reviewer #1: (No Response)

Reviewer #2: (No Response)

2. Is the manuscript technically sound, and do the data support the conclusions?

Reviewer #1: Yes

Reviewer #2: Partly

3. Has the statistical analysis been performed appropriately and rigorously? 

Reviewer #1: Yes

Reviewer #2: No

4. Have the authors made all data underlying the findings in their manuscript fully available?

Reviewer #1: Yes

Reviewer #2: No

5. Is the manuscript presented in an intelligible fashion and written in standard English?

Reviewer #1: Yes

Reviewer #2: Yes

6. Review Comments to the Author

Reviewer #1: Thank you for taking the time revising your manuscript and understanding our queries.

I only have one further query on the choriocapillaris segmentation. We all know that OCTA gives us the unprecedented advantage to quantify choriocapillaris, yet there are many limitations related to its use. In order to obtain the most reliable results we have to control for many confounders. Authors stated that they used a default slab 23-um thick posteriorly to the Bruch using default segmentation.

I have a few queries.

1. Automated segmentation is not enough to correct segmentation errors, manual correction is needed. Could please the Authors repeat the analysis correcting all the segmentation errors?

2. Why did the Authors choose a 23 um thick slab given histologically the CC is reported to be 10-um thick?

3. My other concern is the position of the CC slab. From my understanding the Authors are locating the slab right off the Bruch membrane, right? The selected slab is very close to the RPE and I assume many flow deficits are coming from RPE projection. Did the Authors confirmed the results using a deeper slab?

Reviewer #2: My major concern regarding the method of estimating capillary density was not adequately addressed by the authors response. The author provided several addition sources, however none of these sources justify using a crude average pixel intensity to estimate capillary density. All of the provided sources use image analysis programs to identify vessels and calculate capillary density in a much more reliable way. While many of these sources take multiple images, align them, and average the aligned images together in order to decrease the signal to noise ratio and aid in vessel identification, none of them use average pixel intensity as an estimate of capillary density. There are too many confounding factors that could affect the average intensity of a single image/ROI. While I appreciate that resources may limit the authors access to some of the software or custom analysis tools used by other groups, this type of analysis can be done using free software such as ImageJ.

Furthermore, the figure in which this data is shown (Fig 4) is very unclear. Average pixel intensity is displayed as a percent, but a percent of what? It is also unclear what the error bars represent (SEM vs SD), and there is significant overlap of error bars for all 12 comparisons. . Additionally, the methods state that independent t-tests were used for all statistical analysis, however this type of analysis is inappropriate for this data. This data is comparing multiple dependent variables between two independent groups (ie capillary density of the superior nasal region and capillary density of the inferior nasal region in a single patient are dependent, while the NTG and BRVO groups are independent). Independent t-tests are commonly misused in the analysis of this type of data, and can lead to type I errors especially as the number of dependent variables increases (in this instance there are 12 dependent variables increasing the likelihood of type 1 error). A more appropriate analysis for this data would be one-way multivariate analysis of variance.

Please see the following article on the misuse of t-tests in medical research:

Guangping Liang, Wenlaing Fu, and Kaifa Wang. Analysis of t-test misuses and SPSS operations in medical research papers. Burns Trama. 2019; 7:31.

Other Minor Points:

Line 43: Sentence is grammatically incorrect. Should read “Retinal vein occlusion is a common cause of vision loss…” or “Retinal vein occlusion is one of the most common causes of vision loss…”

Line 76: Need to capitalize Ulsan University Hospital

Line 161: should read “…at the inner plexiform layer, which is the most superficial hyporeflective band.”

7. PLOS authors have the option to publish the peer review history of their article (what does this mean?). If published, this will include your full peer review and any attached files.

Reviewer #1: No

Reviewer #2: No

---

## [Author Response · Author response to Decision Letter 1]

13 Jul 2020

Reviewer #1: Thank you for taking the time revising your manuscript and understanding our queries.

I only have one further query on the choriocapillaris segmentation. We all know that OCTA gives us the unprecedented advantage to quantify choriocapillaris, yet there are many limitations related to its use. In order to obtain the most reliable results we have to control for many confounders. Authors stated that they used a default slab 23-um thick posteriorly to the Bruch using default segmentation.

: Thank you for your pertinent query. As you know, the choroid is divided into five layers. Starting from the retinal side, these include Bruch’s membrane, three vascular layers (the choriocapillaries; Sattler’s layer, which is composed of medium and small arterioles; and Haller’s layer, which includes large arteries and veins), and the suprachoroidea. Torczynski et al. found mean thickness of choriocapillaries was 27.5 μm by histological section after autopsy, and recently, Uji et al. showed the mean diameter of choriocapillary vessels was 22.8 μm by using OCTA. Moreover, choriocapillaries lie directly under Bruch’s membrane, while large vessels of the system lie posterior to the capillaries. We wanted to increase the proportion of choriocapillaries seen; therefore, we set the starting segmentation from under Bruch’s membrane, which is a hyperreflective band by OCTA and has no vascular area in a retinal lesion, and the ending segmentation was set to 23 μm below Bruch’s membrane. In every case, we checked whether the starting segmentation included Bruch’s membrane, or was over Bruch’s membrane not including the RPE layer. If there was such a case, then we manually corrected the starting segmentation not to include Bruch’s membrane or the area over it. 

I have a few queries.

1. Automated segmentation is not enough to correct segmentation errors, manual correction is needed. Could please the Authors repeat the analysis correcting all the segmentation errors?

: Thank you for your concern. We provided an answer to this query in the above paragraph. 

2. Why did the Authors choose a 23 um thick slab given histologically the CC is reported to be 10-um thick?

: We appreciate the question. We provided an answer to this in the above paragraph.

3. My other concern is the position of the CC slab. From my understanding the Authors are locating the slab right off the Bruch membrane, right? The selected slab is very close to the RPE and I assume many flow deficits are coming from RPE projection. Did the Authors confirmed the results using a deeper slab?

: We further described our methods in the above paragraph.

Reference)

1. Joanna Kur 1, Eric A Newman, Tailoi Chan-Ling Cellular and Physiological Mechanisms Underlying Blood Flow Regulation in the Retina and Choroid in Health and Disease. Prog Retin Eye Res 2012 Sep;31(5):377-406

2. E Torczynski, M O Tso. The Architecture of the Choriocapillaris at the Posterior Pole. Am J Ophthalmol 1976 Apr;81(4):428-40

3. Reference 22 in manuscript

Reviewer #2: My major concern regarding the method of estimating capillary density was not adequately addressed by the authors response. The author provided several addition sources, however none of these sources justify using a crude average pixel intensity to estimate capillary density. All of the provided sources use image analysis programs to identify vessels and calculate capillary density in a much more reliable way. While many of these sources take multiple images, align them, and average the aligned images together in order to decrease the signal to noise ratio and aid in vessel identification, none of them use average pixel intensity as an estimate of capillary density. There are too many confounding factors that could affect the average intensity of a single image/ROI. While I appreciate that resources may limit the authors access to some of the software or custom analysis tools used by other groups, this type of analysis can be done using free software such as ImageJ.

Furthermore, the figure in which this data is shown (Fig 4) is very unclear. Average pixel intensity is displayed as a percent, but a percent of what? It is also unclear what the error bars represent (SEM vs SD), and there is significant overlap of error bars for all 12 comparisons. Additionally, the methods state that independent t-tests were used for all statistical analysis, however this type of analysis is inappropriate for this data. This data is comparing multiple dependent variables between two independent groups (ie capillary density of the superior nasal region and capillary density of the inferior nasal region in a single patient are dependent, while the NTG and BRVO groups are independent). Independent t-tests are commonly misused in the analysis of this type of data, and can lead to type I errors especially as the number of dependent variables increases (in this instance there are 12 dependent variables increasing the likelihood of type 1 error). A more appropriate analysis for this data would be one-way multivariate analysis of variance.

Please see the following article on the misuse of t-tests in medical research:

Guangping Liang, Wenlaing Fu, and Kaifa Wang. Analysis of t-test misuses and SPSS operations in medical research papers. Burns Trama. 2019; 7:31.

: We sincerely regret that our previous response did not fully address your major concern. 

First, we must explain i-solution software. i-solution is image analysis software that is similar to image J. However, I believe it is more convenient compared to image J, because it is a well-designed platform to analyze images. i-solution software has been used in the basic research field for detection of cell number, counting axons of neurons, and in the heavy material field to calculate exact length and exact area from acquired images. Uses of i-solution in basic research were cited in the manuscript. 

Moreover, our procedure of image analysis was very similar to that of the Mansoori study, which was provided previously. To summarize processing, there are 3 main stages. The first stage is extraction of the region of interest (ROI) around the ONH. The second stage is detection and suppression of thick vessels. In this stage, there was some difference between our study and that of Mansoori. They used the Bar-Selectiye combination of shifted filter responses method, but in our study, we excluded large vessels such as the superior and inferior retinal artery and vein one-by-one using sub mode (deletion), a built-in program of i-solution. Even though there was some worry about not fully suppressing large vessels using this process, in the newly provided image (Figure 3) we can see that almost all large vessels were excluded. The last stage is estimation of peripapillary vessel density, which is the ideal measure of capillaries per unit area. The desired image is the binary image, in which white pixels represent capillaries (without large vessels) and black pixels represent non-vessels. The peripapillary vessel density was calculated using the following formula: peripapillary vessel density = (Nw / A )× 100, where Nw, represents the number of white pixels and A represents the area of the selected image sector. As both the numerator and the denominator are pixel counts, the peripapillary vessel density is reported between 0% and 100%. We included these processes and the figures in the methods section of the manuscript.

The error bar in Figure 4 represents standard deviation (SD), and we included these contents in the Figure 4 legend.

We purposefully discussed the statistical analysis of Figure 4 with the statistician in the medical information and research center of our hospital. We finally concluded that your suggestion was more reasonable, and we rechecked our data with one-way multivariate analysis of variance (MANOVA). After this, there were no significant changes in p-values, and we have included the results of MANOVA below. So, we needed not change Figure 4, and we concluded using figure 4 continuously instead of table, because in the figure we easily see the differences at a glance. 

Other Minor Points:

Line 43: Sentence is grammatically incorrect. Should read “Retinal vein occlusion is a common cause of vision loss…” or “Retinal vein occlusion is one of the most common causes of vision loss…”

: We changed this sentence as you recommended and double-checked it using Editage.

Line 76: Need to capitalize Ulsan University Hospital

: We changed this sentence as you recommended and double-checked it using Editage. 

Line 161: should read “…at the inner plexiform layer, which is the most superficial hyporeflective band.”

: We changed this sentence as you recommended and double-checked it using Editage.

---

## [Decision Letter · Decision Letter 2]

17 Aug 2020

PONE-D-20-01336R2

Comparison of lamina cribrosa properties and the peripapillary vessel density between branch retinal vein occlusion and normal-tension glaucoma

PLOS ONE

Dear Dr. Lee,

Thank you for submitting your manuscript to PLOS ONE. After careful consideration, we feel that it has merit but does not fully meet PLOS ONE’s publication criteria as it currently stands. Therefore, we invite you to submit a revised version of the manuscript that addresses the points raised during the review process.

A reviewer has recommended minor revision that can be easily performed by incorporating changes in the manuscript.

We look forward to receiving your revised manuscript.

Kind regards,

Sanjoy Bhattacharya

Academic Editor

PLOS ONE

Reviewers' comments:

Reviewer's Responses to Questions

**Comments to the Author**

1. If the authors have adequately addressed your comments raised in a previous round of review and you feel that this manuscript is now acceptable for publication, you may indicate that here to bypass the “Comments to the Author” section, enter your conflict of interest statement in the “Confidential to Editor” section, and submit your "Accept" recommendation.

Reviewer #1: All comments have been addressed

Reviewer #2: All comments have been addressed

2. Is the manuscript technically sound, and do the data support the conclusions?

Reviewer #1: Yes

Reviewer #2: Yes

3. Has the statistical analysis been performed appropriately and rigorously? 

Reviewer #1: Yes

Reviewer #2: Yes

4. Have the authors made all data underlying the findings in their manuscript fully available?

Reviewer #1: Yes

Reviewer #2: No

5. Is the manuscript presented in an intelligible fashion and written in standard English?

Reviewer #1: Yes

Reviewer #2: Yes

6. Review Comments to the Author

Reviewer #1: Thank you for the time spent revising my comments.

I have a main concern related to the choriocapillaris quantification.

The Authors mentioned that as published recently, Uji et al showed the mean diameter of choriocapillary vessels was 22.8 μm by using OCTA. These findings are simply describing the mean diameter of the vessels, not referring to the location of CC and thickness.

More recently, Byon et al published on the most repeatible and reliable CC slab in quantitative studies. I would suggest to revise the following manuscript and consider improving the methodology.

Byon I, Alagorie AR, Ji Y, Su L, Sadda SR. Optimizing the Repeatability of Choriocapillaris Flow Deficit Measurement from Optical Coherence Tomography Angiography [published online ahead of print, 2020 May 23]. Am J Ophthalmol. 2020;S0002-9394(20)30266-X. doi:10.1016/j.ajo.2020.05.027

Reviewer #2: Thank you for further clarifying your analysis of vessel density. Prior descriptions of the analysis methods did not make it clear that images were converted from gray scale to a binary image. Given this clarification, I agree that the method of analysis used here is appropriate. My only suggestion would be to include an example of the final binary image in figure 3, as the images shown in are still in gray scale. Additionally, the authors re-analyzed this data using one-way multivariate analysis of variance and addressed my concerns regarding appropriate statistical analysis.

7. PLOS authors have the option to publish the peer review history of their article (what does this mean?). If published, this will include your full peer review and any attached files.

Reviewer #1: No

Reviewer #2: No

---

## [Author Response · Author response to Decision Letter 2]

14 Sep 2020

Reviewer #1: Thank you for the time spent revising my comments.

I have a main concern related to the choriocapillaris quantification.

The Authors mentioned that as published recently, Uji et al showed the mean diameter of choriocapillary vessels was 22.8 μm by using OCTA. These findings are simply describing the mean diameter of the vessels, not referring to the location of CC and thickness.

More recently, Byon et al published on the most repeatible and reliable CC slab in quantitative studies. I would suggest to revise the following manuscript and consider improving the methodology.

Byon I, Alagorie AR, Ji Y, Su L, Sadda SR. Optimizing the Repeatability of Choriocapillaris Flow Deficit Measurement from Optical Coherence Tomography Angiography [published online ahead of print, 2020 May 23]. Am J Ophthalmol. 2020;S0002-9394(20)30266-X. doi:10.1016/j.ajo.2020.05.027

:Thank you for your pertinent suggestion. Accordingly, we rechecked the choriocapillaris with the method used by Byon et al, that is,a slab of choriocapillaris was identified as a 10-μm thick slab offset of 21μm below the instrument generating the retinal pigment epithelial band and have added this information in the manuscript. There was no significant difference between the results obtained by the new method and that obtained by the previously used result. 

We think that one of the reasons for the above result was that we manually corrected the starting segmentation to not include the Bruch’s membrane or the area over as was done in our previous interpretation.

We have presented this new result as Figure 4. 

Reviewer #2: Thank you for further clarifying your analysis of vessel density. Prior descriptions of the analysis methods did not make it clear that images were converted from gray scale to a binary image. Given this clarification, I agree that the method of analysis used here is appropriate. My only suggestion would be to include an example of the final binary image in figure 3, as the images shown in are still in gray scale. Additionally, the authors re-analyzed this data using one-way multivariate analysis of variance and addressed my concerns regarding appropriate statistical analysis.

:Thank you for your comment. Accordingly, we have added the final binary image in Figure 3 as part D.

---

## [Editor Report · Decision Letter 3]

21 Sep 2020

Comparison of lamina cribrosa properties and the peripapillary vessel density between branch retinal vein occlusion and normal-tension glaucoma

PONE-D-20-01336R3

Dear Dr.Lee,

We’re pleased to inform you that your manuscript has been judged scientifically suitable for publication and will be formally accepted for publication once it meets all outstanding technical requirements.

Kind regards,

Sanjoy Bhattacharya

Academic Editor

PLOS ONE
---

## [Editor Report · Acceptance letter]

24 Sep 2020

PONE-D-20-01336R3 

Comparison of lamina cribrosa properties and the peripapillary vessel density between branch retinal vein occlusion and normal-tension glaucoma 

Dear Dr. Lee:

I'm pleased to inform you that your manuscript has been deemed suitable for publication in PLOS ONE. Congratulations! Your manuscript is now with our production department. 

Kind regards, 

on behalf of

Dr. Sanjoy Bhattacharya 

Academic Editor

PLOS ONE